# Identification of nanomolar adenosine A$_{2A}$ receptor ligands using reinforcement learning and structure-based drug design

Morgan Thomas[1], Pierre G. Matricon [2], Robert J. Gillespie[2], Maja Napiórkowska [2], Hannah Neale[2], Jonathan S. Mason[2], Jason Brown[2], Kaan Harwood[2], Charlotte Fieldhouse[2], Nigel A. Swain[2], Tian Geng [2], Noel M. O'Boyle [2], Francesca Deflorian[2] ✉, Andreas Bender [1,3,4] ✉ & Chris de Graaf[2,5] ✉

Generative chemical language models (CLMs) have demonstrated success in learning language-based molecular representations for de novo drug design. Here, we integrate structure-based drug design (SBDD) principles with CLMs to go from protein structure to novel small-molecule ligands, without a priori knowledge of ligand chemistry. Using Augmented Hill-Climb, we successfully optimise multiple objectives within a practical timeframe, including protein-ligand complementarity. Resulting de novo molecules contain known or promising adenosine A$_{2A}$ receptor ligand chemistry that is not available in commercial vendor libraries, accessing commercially novel areas of chemical space. Experimental validation demonstrates a binding hit rate of 88%, with 50% having confirmed functional activity, including three nanomolar ligands and two novel chemotypes. The two strongest binders are co-crystallised with the A$_{2A}$ receptor, revealing their binding mechanisms that can be used to inform future iterations of structure-based de novo design, closing the AI SBDD loop.

The discovery and design of small molecules that meet specific target endpoints in drug discovery is a formidable challenge. Traditionally, structure-based hit discovery involves computationally screening commercially available or proprietary libraries via docking into the binding pocket, with expected hit rates between 0.02% and 34.8%[1]. However, the rapid expansion of virtual libraries[2] renders brute force search practically intractable without the use of artificial intelligence (AI) to narrow the search space[3,4]. Generative AI, on the other hand, can learn larger chemical spaces[5] - up to 1000-fold greater[6] - which are also faster to traverse due to their implicit nature, promising to significantly enhance drug discovery efficiency[7].

Chemical language models (CLMs)[8–10], are a particular class of neural networks trained on datasets using a chemical language representation such as SMILES[11]. These models can be combined with optimisation techniques such as reinforcement learning (RL)[12–14] to guide de novo molecule generation towards specific endpoints by a reward signal computed by molecule evaluation functions or scoring functions. Furthermore, these models have shown consistent state-of-the-art performance on several benchmarks[14–18] and are the most used models for generative molecular design[19]. These models are also one of the most extensively experimentally validated[20–25], as also recently reviewed[26]. However, none of the experimental validations have used CLMs for protein structure-based drug design (SBDD).

[1]Centre for Molecular Informatics, Department of Chemistry, University of Cambridge, Cambridge, UK. [2]Nxera Pharma, Steinmetz Building, Granta Park, Great Abington, Cambridge, UK. [3]College of Medicine and Health Sciences, Khalifa University of Science and Technology, Abu Dhabi, United Arab Emirates. [4]STAR-UBB Institute, Babeş-Bolyai University, Cluj-Napoca, Romania. [5]Present address: Structure Therapeutics, 601 Gateway Blvd, San Francisco, CA, USA. ✉e-mail: Francesca.Deflorian@nxera.life; andreas.bender@ku.ac.ae; chrisdgrf@gmail.com

The predominant benefit of structure-based over ligand-based drug design is the ability to explore novel chemical spaces that are complementary to protein structure, unrestricted by known ligand space. Retrospective analyses have shown additional benefits, including increased diversity of known ligand chemistry rediscovered[27] compared to the use of ligand-based approaches, which can exacerbate generative model failure modes[28]. However, the practical implementation of SBDD often comes at the expense of increased computational run time[27,29]. Therefore, the use of SBDD as scoring functions in RL is limited by the learning efficiency of the algorithm (i.e., how many samples are required), a topic which is becoming of increasing importance[16,30–33].

In this work, we use a CLM[8] in combination with a sample-efficient RL algorithm, Augmented Hill-Climb (AHC)[30], to design putative adenosine $A_{2A}$ receptor ligands. The overall workflow is demonstrated in Fig. 1. The increased efficiency and hence reduced run-time enable extensive exploration of the effects of protein co-crystal structure and scoring function protocol on de novo chemistry. We further synthesize nine proposed molecules and test bioactivity with respect to both receptor binding and functional activity. The two most potent nanomolar compounds are co-crystallised with the $A_{2A}$ receptor to improve our understanding of receptor binding.

## Results

### Design of putative $A_{2A}$ receptor ligands

To generate de novo molecules with optimal properties, we used a CLM trained using RL. First, a recurrent neural network was trained at next token prediction using maximum likelihood estimation on 189,238 SMILES string extracted from the ChEMBL database[34], constituting our CLM. Second, this CLM underwent further training using RL. In RL, the CLM equates to a policy that decides which action (next

token) to take given the current state (previously observed tokens), such as to learn how to maximise a reward. We used the AHC[30] algorithm for its improved learning efficiency compared to baseline algorithms[12,35–37] which fine-tunes the CLM to maximise a reward bound between [0, 1]. Note that a copy of the pre-trained CLM is kept during RL and is used as a prior policy to regularise learning and maintain the chemical principles initially learned.

AHC was then used to train the CLM to generate molecules optimal against each of the seven $A_{2A}$ receptor crystal structures over the course of sampling 12,800 de novo molecules per structure. The reward maximised was formulated to reflect molecular desirability by combining the predicted protein complementarity according to the GlideSP docking score and four secondary objectives to encourage more favourable drug-like properties. These secondary objectives included synthesisability[38], predicted logP, hydrogen bond donor count, and the maximum number of consecutive rotatable bonds present, thus presenting a more realistic and challenging multi-objective optimisation problem.

Figure 2 demonstrates that AHC successfully generated molecules with favourable docking scores for the respective $A_{2A}$ receptor structure while maintaining secondary objectives within desirable ranges, highlighting the powerful multi-objective optimisation ability of AHC within a restricted budget of just 12,800 molecules. To further understand how the optimisation of the structure-based objective influenced de novo chemistry, we compared the generated molecules to known $A_{2A}$ receptor ligands (sourced from GPCRBench[39] and updated with newer data from ChEMBL29[34] and Reaxys), classified by chemotype. Note that $A_{2A}$ receptor ligand chemotypes were defined manually by in-house project teams utilising X-ray and docked co-structures as per GPCRBench[39]. Figure 3 shows that each crystal structure led to rediscovery of approximately 25 unique chemotypes

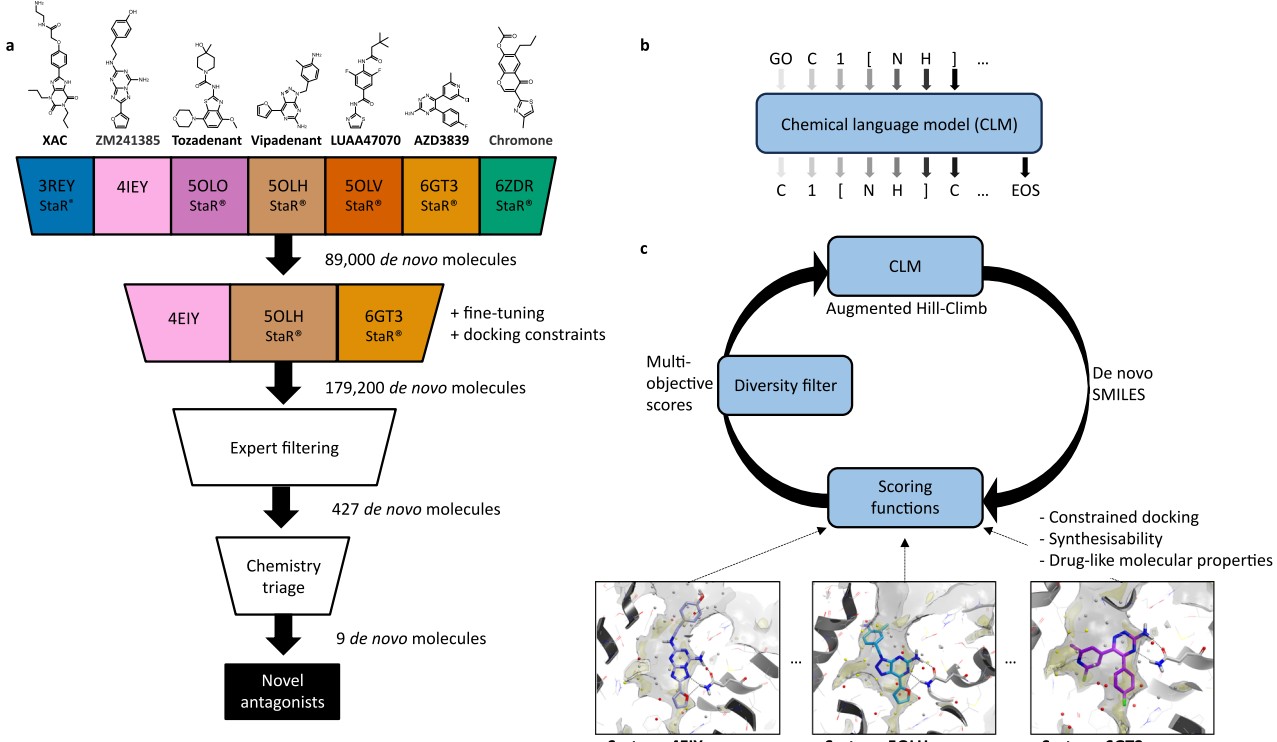

**Fig. 1 | Overview of the workflow presented here. a** The computational pipeline starting with crystal structures and yielding novel $A_{2A}$ receptor ligands. The training in (**c**) is repeated over multiple crystal structures used for structure-based design with different constraints. The crystallographic structures 4EIY, 5OLH, and 6GT3 are demonstrative, all crystallographic structures used in this study are shown in Supplementary Fig. 1. **b** The Chemical Language Model (CLM) is pre-trained by next token prediction on SMILES strings, learning to generate SMILES strings de novo. **c** The CLM is then further trained using reinforcement learning (RL) with Augmented Hill-Climb (AHC) to maximise the multi-objective scores of de novo molecules.

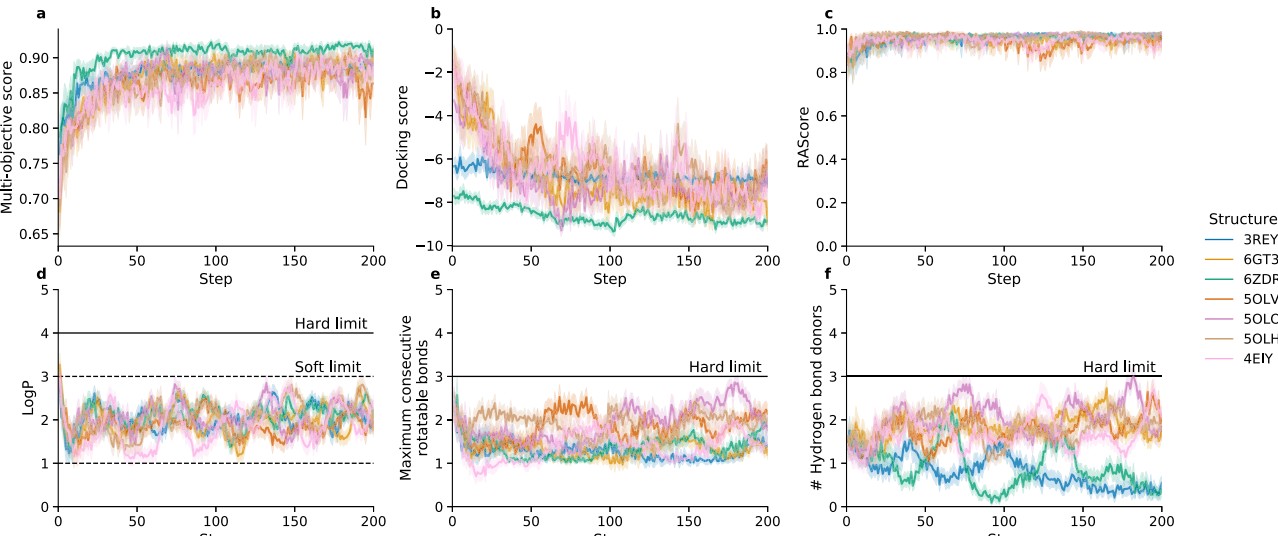

**Fig. 2 | Optimisation of the structure-based objective repeated over seven different experiments each with a different A_2A receptor structure.**
**a** Optimisation of the overall multi-objective, an arithmetic mean of the following five different components: (**b**) the docking score against the respective structure (lower is better), (**c**) the RAScore predicting the synthesisability of molecules (higher is better), (**d**) predicted logP, (**e**) maximum number of consecutive rotatable bonds in the molecule, and **f** the number of hydrogen bond donors. All scores are transformed to a range between zero and one. Error bands (**a**–**e**) show the 95% confidence interval. The last three objectives are annotated "soft limit" or "hard limit". If the respective property is within the soft limit, it is given a score of 1, if it is beyond the hard limit, it is given a score of 0. Between the soft and hard limit, it is linearly scored between one to zero from the soft to hard limit. In **e** and **f** the soft limit is the same as the hard limit. Source data are provided as a Source Data file.

known to be present in A_2A ligands, except for 6ZDR, which led to the rediscovery of approximately 18. This corresponded to a range of 2000 to 4000 molecules per experiment with known A_2A chemotypes out of the 12,800 (~15–30%), indicating alignment between the structure-based objective and known A_2A ligands. Interestingly, only one to two known active molecules were rediscovered per experiment meaning that almost all de novo molecules are novel, providing a significant enrichment around known A_2A ligand chemistry. The number and type of chemotypes rediscovered varied depending on the structure used; for example, only the experiment with the 5OLV structure rediscovers the 2c6 chemotype. All chemotypes rediscovered are detailed in Supplementary Fig. 2.

Another important consideration is novelty with respect to commercial vendor libraries. Figure 3e shows that approximately 10,000 molecules are novel with respect to a combination of MolPort, ChemSpace and Aldrich (a total of ~11.7 M purchasable compounds at the time of this study). Of these, about 2000 to 4000 contained known A_2A chemotypes, highlighting promising compounds that traditional virtual screening methods would not discovery, thus delving into novel chemical space.

Overall, structure-based drug design objectives can be efficiently optimized with a CLM and AHC, resulting in the rediscovery of commercially novel but relevant ligand chemistry, dependent on the choice of structure used.

Despite clear enrichment of known A_2A receptor ligand chemistry, the majority of molecules do not possess known chemotypes. However, chemotypes that have multiple corresponding de novo molecules with high scores could represent promising novel chemotypes compared to known A_2A receptor ligands. To identify promising unknown chemotypes, molecules with a GlideSP score less than −10 were pooled across all experiments and unknown chemotypes extracted. Note that of the 10,363 de novo molecules with a GlideSP score less than −10, only 3101 possessed unknown chemotypes, indicating that at this threshold, known A_2A chemotypes are much more prevalent. The unknown chemotypes were clustered using the Tanimoto similarity (using ECFP4 fingerprints) of the Bemis-Murcko scaffolds with a similarity cut-off of 0.2 and the

maximum common substructure of each cluster was identified resulting in 65 chemotypes derived from a cluster containing at least 3 example molecules (see Supplementary Fig. 3). Many of these chemotypes still display functional groups important for interacting with N253[6.55] (Ballesteros-Weinstein generic residue numbers are shown in superscript), and therefore are of interest to validate as potentially novel ligand series.

As previously established, the protein structure used influences the resulting de novo chemistry. We also tested further modifications, such as the effect of fine-tuning the CLM on an A_2A receptor ligand dataset prior to applying AHC. While this initially increased the number of unique chemotypes rediscovered, the number decreased over time due to the diversity filter penalising repeated areas of chemical space. By the end of the AHC-based optimisation the non-fine-tuned CLM generated a higher proportion of molecules with known A_2A chemotypes (see Supplementary Fig. 4), indicating that the effect of fine-tuning the CLM is short-lived as RL updates the model parameters.

We also investigated the effect of different docking constraints, such as enforcing a docked pose to interact with specific sub-pockets based on pharmacophoric features. The default constraints used for all seven experiments required either a hydrogen bond donor or acceptor interaction with N253[6.55], and the occupation by a lipophilic moiety of either of the sub-pockets II and III (see Fig. 4)[40,41]. These constraints were made stricter on a subset of three crystal structures (4EIY, 5OLH and 6GT3) selected to balance rediscovery of A_2A chemotypes and novelty. Figure 4 shows that stricter docking constraints lead to a greater rediscovery of unique chemotypes. However, it is not predictable whether stricter constraints result in more de novo molecules with known A_2A chemotypes. Enforcing occupation of both lipophilic sub-pockets decreases the number of total molecules with A_2A chemotypes, while requiring both a hydrogen bond donor and acceptor interaction with N253[6.55] increases this number. Experimenting with different docking constraints resulted in the rediscovery of four additional chemotypes compared to the default constraints. Moreover, we observed an increase in the quality of predicted poses of the de novo molecules, although this is more challenging to measure quantitatively. The resultant de novo molecules from these additional

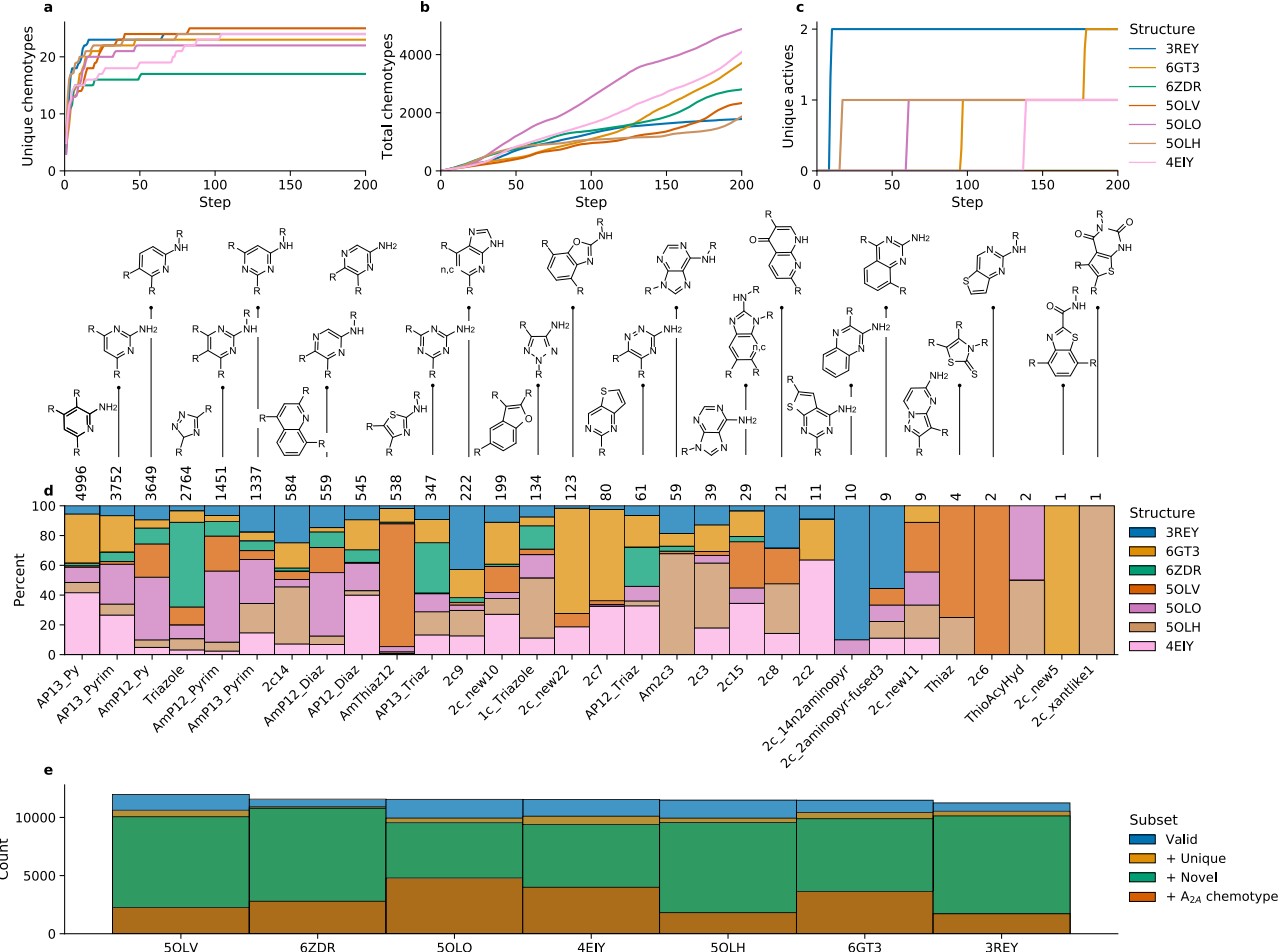

**Fig. 3 | Chemistry assessment of de novo molecules with respect to known A$_{2A}$ ligands and their chemotypes. a** Number of unique A$_{2A}$ chemotypes identified during AHC optimisation of the objective. **b** Total number of unique molecules generated that possess known A$_{2A}$ chemotypes during AHC optimisation. **c** Number of unique known A$_{2A}$ active molecules that were rediscovered during AHC optimisation. **d** Number of molecules identified containing a particular chemotype, as well as the percentage breakdown per structure used. **e** Proportion of de novo molecules per experiment that are valid; valid and unique; valid, unique, and novel with respect to common commercial libraries at the time of this study; and finally valid, unique, novel, and possessing a known A$_{2A}$ chemotype. Source data are provided as a Source Data file.

experiments were pooled together with the previous seven experiments for further analysis, filtering, and selection.

## Selection of de novo ligands for experimental validation

To select candidates for synthesis and testing, the de novo proposals were filtered down. First, the top 1000 unique molecules from each experiment, based on the multi-objective desirability score, were visually inspected. From this, 427 molecules of interest were inspected and selected based on the predicted binding pose. Selection criteria included: (i) H-bond interaction with N253$^{6.55}$; (ii) interaction with at least two of the following three lipophilic hotspots II, III, IV, displacing energetically unfavourable unhappy water molecules located in these binding site regions (Fig. 4). Analysis of these 427 compounds revealed that 6 were already known ligands (i.e. 98.6% were novel with respect to known ligands) and 8 were available for purchase in vendor libraries (i.e. 98.1% were novel with respect to commercial libraries). The majority (75.6%) contained known A$_{2A}$ chemotypes, providing confidence that the chemistry was novel yet relevant. To estimate synthetic feasibility, AiZynthFinder[42], which has a reported accuracy of up to 80%[43], was used to predict synthetic routes. 71% of the 427 molecules were predicted to be synthesisable via a proposed synthetic route.

To complement this, WaterFLAP[44] was used to calculate the pseudo-apo water network for each A$_{2A}$ structure binding site as well as perturbation of the pseudo-apo water network upon ligand binding for the 427 molecules and predicted binding poses. This helped to de-prioritise molecules predicted to trap water molecules unfavourably in the lipophilic pockets without a sufficient supporting water network or hydrogen bond donors/acceptors. Based on these analyses, a panel of 41 predicted synthesisable de novo molecules was proposed for synthesis, of which 9 were synthesised for experimental validation based on synthetic feasibility and diversity. Synthetic chemistry routes of this triaged set of compounds were defined based on organic chemistry knowledge and established synthesis protocols. Chemical diversity-based selection was guided by ECFP4 fingerprint clustering. The final 9 molecules are shown in Fig. 5 with their corresponding predicted binding pose in Fig. 6. Four of these molecules contain known A$_{2A}$ chemotypes, while the remainder contain potentially novel bioactive chemotypes. None of the compounds were available for purchase from a commercial library, and none had a Tanimoto similarity (using ECFP4 fingerprints) greater than 0.6 to any compound in the training dataset.

## Experimental validation and characterisation of de novo ligands

To confirm the ability of the synthesized compounds to bind to the A$_{2A}$ receptor, the compounds were tested for their ability to displace [$^3$H]-ZM241385. ZM241385 was previously demonstrated to be an inverse

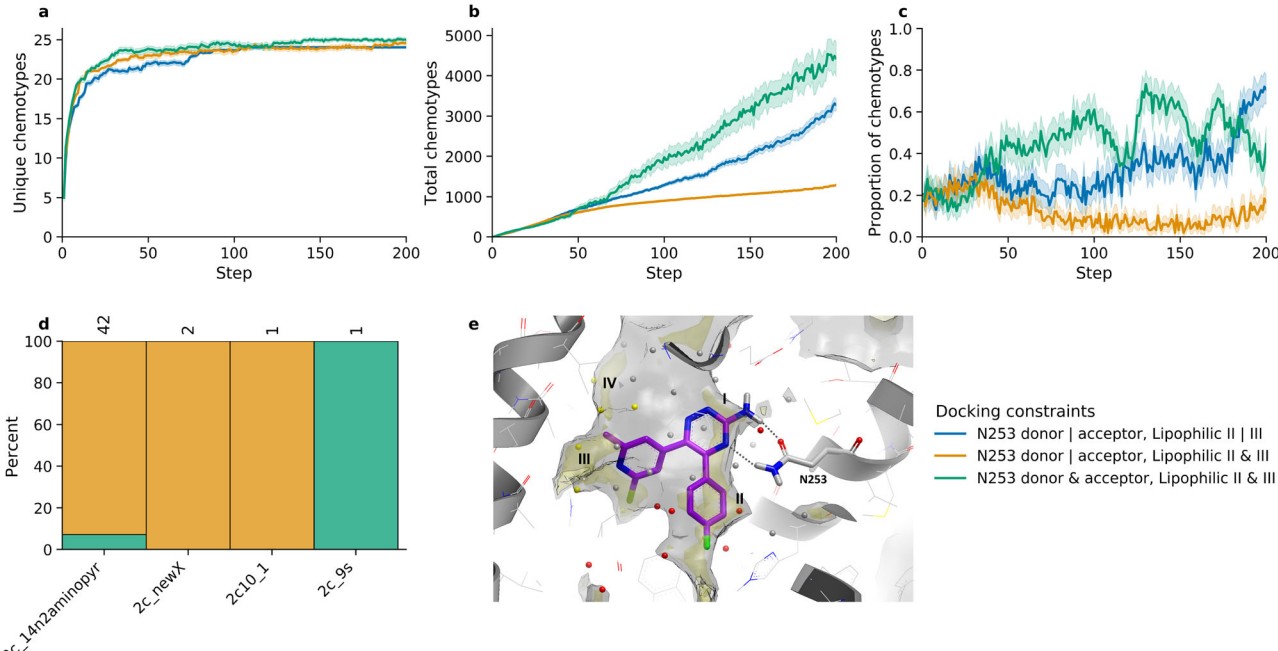

**Fig. 4 | The effect of additional docking constraints on a subset of three crystal structures 4EIY, 5OLH and 6GT3. a** Number of unique chemotypes identified according to docking constraints used. **b** Total number of unique molecules generated that possess known $A_{2A}$ chemotypes according to docking constraints used. **c** Proportion of molecules containing $A_{2A}$ chemotypes per step during optimisation according to docking constraints used. Error bands (**a**–**c**) show the 95% confidence interval. **d** Additional $A_{2A}$ chemotypes identified with the use of stricter docking constraints. **e** Binding site representation of the crystal structure 6GT3. The receptor is shown in grey cartoon with residue side chains in wireframe representation with grey carbons, whereas N253[6.55] and co-crystallised ligand in stick representation, with the ligand carbon atoms coloured in purple. Hydrogen bonds to N253[6.55] are shown with grey dashed lines. GRID maps are shown as transparent solids, with the pocket surface, in terms of how close a ligand carbon atom can approach, contoured by a CH3 methyl probe at 1 kcal/mol (in grey) and lipophilic hotspots sub-pockets contoured by a sp2 CH probe (C1=) probe at −2.8 kcal/mol (in yellow). The lipophilic hot spots annotated in sub-pockets I to III are respectively occupied by the triazine, the 4-fluorophenyl and the substituted pyridyl rings of the ligand; sub-pocket IV contains another lipophilic hot spot located above pocket III. WaterFLAP water networks calculated on the pseudo-apo binding site (with the ligand removed) are shown as spheres and colour-coded by relative energetic scoring in red when predicted free energy (ΔG) is higher than 3.0 kcal/mol, yellow when ΔG is between 1.5 and 3.0 kcal/mol, grey if ΔG is between −2.0 and 1.5 kcal/mol, and blue when ΔG <−2.0 kcal/mol. Source data are provided as a Source Data file.

agonist at the $A_{2A}$ receptor[41]. All compounds competed with [³H]-ZM241385, with some compounds demonstrating a pKi >7.3 (Table 1, Supplementary Fig. 5).

To characterise the agonist pharmacology of the synthesised compounds, the ability of the compounds to cause accumulation of cAMP in response to the ligands was measured. Compound 5 increased cAMP accumulation in the $A_{2A}$ CHO-K1 (pEC$_{50}$ 7.3); however, maximum cAMP accumulation levels were lower than that of NECA (E$_{max}$ 65%). This suggests that Compound 5 is an $A_{2A}$ receptor partial agonist.

Overexpression of $A_{2A}$ in CHO-K1 resulted in elevated cAMP accumulation in the absence of a ligand, suggesting the $A_{2A}$ receptor has high constitutive activity. Compounds 7, and 9 caused inhibition of the constitutive activity (Table 1, Fig. 7). These compounds are consequently considered inverse agonists of the $A_{2A}$ receptor.

To characterise the antagonist pharmacology of the synthesised compounds at both the $A_{2A}$ and $A_{2B}$ receptor, the ability to inhibit NECA-induced cAMP accumulation was measured. Pharmacological characterisation of the compounds at the $A_{2B}$ receptor, showed no inhibition of NECA-induced cAMP accumulation at the concentrations tested. The previously mentioned Compounds 7 and 9 caused inhibition of cAMP at the $A_{2A}$ receptor, in the presence of NECA, further suggesting inverse agonism. Compound 4 (isomers 1 and 2) showed inhibition of the EC$_{80}$ of NECA at the $A_{2A}$ receptor, but did not inhibit the constitutive activity, suggesting that these compounds could be neutral antagonists.

Compounds 1, 2, 3, 6, and 8 did not cause any modulation of cAMP accumulation or inhibition of NECA-induced cAMP at the concentrations tested, at either the $A_{2A}$ or $A_{2B}$ receptor.

The two strongest binders, Compounds 7 and 9 were successfully co-crystallised with the $A_{2A}$ receptor to reveal their binding modes, as shown in Fig. 8. GRID was used for physicochemical analysis of the binding sites of the two crystallographic structures, highlighting the same lipophilic sub-pockets I to IV (shown in Fig. 4e). Compound 9 binds in a very similar orientation to that predicted by molecular docking to $A_{2A}$ receptor structure 4EIY binding site, with the amino pyrrolopyrimidine core anchored in sub-pocket I by the hydrogen bond interactions to N253[6.55] and E169[ECL2], and the furan ring located in the lipophilic hotspot II. The pyridine moiety of Compound 9 is located as predicted in the lipophilic sub-pocket III. In contrast, the bioactive orientation of Compound 7 is partially different from the predicted docking pose to $A_{2A}$ receptor structure 6GT3. The aminotriazine core and furan ring of Compound 7 are located as expected and as predicted by docking in sub-pockets I and II of the binding site, with the pyridine moiety between sub-pocket III and IV (Supplementary Fig. 6). However, the imidazole moiety is oriented upwards in sub-pocket IV, and not as predicted downwards in sub-pocket III, leaving sub-pocket III filled with water molecules. In both structures, sub-pocket III contains a small water network of four crystallographic waters not displaced by the ligands, and engaged by hydrogen bond interactions with each other, the receptor residues (A59[2.57], I80[3.28], V84[3.32], and H278[7.43]), and the ligands heteroatoms facing sub-pocket III. We note that the second-best scored pose of 7 to 6GT3 $A_{2A}$ receptor binding site showed the imidazole ring pointing upwards in sub-pocket IV as in the co-crystallised structure.

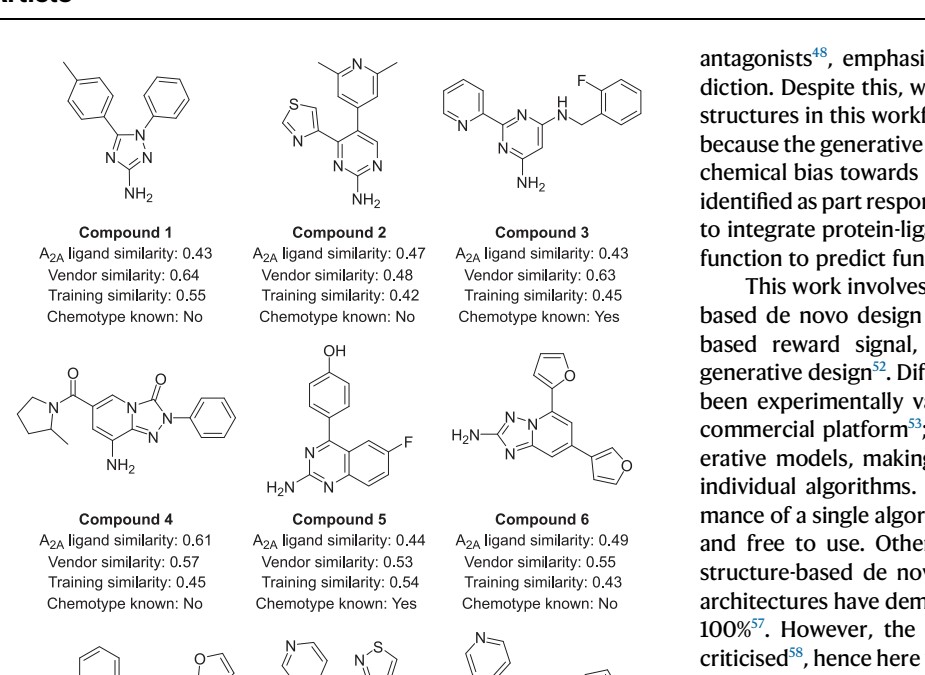

**Compound 1**
A$_{2A}$ ligand similarity: 0.43
Vendor similarity: 0.64
Training similarity: 0.55
Chemotype known: No

**Compound 2**
A$_{2A}$ ligand similarity: 0.47
Vendor similarity: 0.48
Training similarity: 0.42
Chemotype known: No

**Compound 3**
A$_{2A}$ ligand similarity: 0.43
Vendor similarity: 0.63
Training similarity: 0.45
Chemotype known: Yes

**Compound 4**
A$_{2A}$ ligand similarity: 0.61
Vendor similarity: 0.57
Training similarity: 0.45
Chemotype known: No

**Compound 5**
A$_{2A}$ ligand similarity: 0.44
Vendor similarity: 0.53
Training similarity: 0.54
Chemotype known: Yes

**Compound 6**
A$_{2A}$ ligand similarity: 0.49
Vendor similarity: 0.55
Training similarity: 0.43
Chemotype known: No

**Compound 7**
A$_{2A}$ ligand similarity: 0.41
Vendor similarity: 0.50
Training similarity: 0.42
Chemotype known: Yes

**Compound 8**
A$_{2A}$ ligand similarity: 0.34
Vendor similarity: 0.64
Training similarity: 0.38
Chemotype known: No

**Compound 9**
A$_{2A}$ ligand similarity: 0.43
Vendor similarity: 0.46
Training similarity: 0.42
Chemotype known: No

**Fig. 5 | Molecules that underwent synthesis and validation.** Each molecule is annotated with the highest similarity to a known A$_{2A}$ ligand, the highest similarity to a commercial vendor library compound, the highest similarity to a molecule present in the generative model training dataset, and whether the chemotype is a known A$_{2A}$ chemotype. Source data are provided as a Source Data file.

## Discussion

The workflow presented here has led to the discovery of promising, potent antagonists for the A$_{2A}$ receptor using CLMs, RL, and SBDD. Compared to virtual screening of commercial vendor libraries, this approach offers key benefits with regard to chemical novelty and target chemistry enrichment. Approximately 16-30% of de novo molecules per experiment contain commercially novel molecules containing known target chemotypes, which are therefore of immediate interest from a practical drug discovery perspective. This enrichment in high-scoring, desirable compounds led to a hit rate of 88% (8/9) based on binding. This hit rate is higher than previously reported virtual screening efforts in the hunt for A$_{2A}$ ligands, which were between 3% and 64%[45–50]. This is a considerable improvement, especially given that identifying novel A$_{2A}$ ligands is increasingly difficult due to exhaustive exploration of A$_{2A}$ ligand chemical space in the scientific domain. This is highlighted by Lenselink et al., who found only 2 active molecules out of 71 selected when enforcing high degrees of novelty[49].

We note that although 3/4 molecules synthesised functionally inactivated the A$_{2A}$ receptor, one activated the receptor as a partial agonist despite an attempt to bias ligand chemistry using inactive state A$_{2A}$ receptor complexes. However, active and inactive state binding site conformations are generally quite similar, and ligands typically have some affinity for both conformations[45] therefore, it is not surprising to identify ligands with alternative functional effects than expected (especially considering imperfect binding mode prediction with docking). Moreover, a previous study by Rodríguez et al. using activated state A$_{2A}$ receptors in the hunt for agonists only identified

antagonists[48], emphasising the challenge of functional activity prediction. Despite this, we cautiously expect that using active state A$_{2A}$ structures in this workflow would bias ligands more towards agonists because the generative model probes novel chemical space, whereas a chemical bias towards antagonists within the commercial library was identified as part responsible by Rodríguez et al. We also identify scope to integrate protein-ligand interaction fingerprints within the scoring function to predict functional activity[51] in future iterations.

This work involves experimental validation for protein structure-based de novo design with CLMs using RL to optimise a structure-based reward signal, constituting structure-implicit goal-directed generative design[52]. Different structure-based generative models have been experimentally validated[26]. Most extensively, the Chemistry42 commercial platform[53]; however, it pools the outcome of many generative models, making it difficult to delineate the performance of individual algorithms. In comparison, we demonstrate clear performance of a single algorithm and workflow which is made open-source and free to use. Other recent examples of prospectively validated structure-based de novo design using alternative generative model architectures have demonstrated hit rates of 22%[54], 92%[55], 0.83%[56] and 100%[57]. However, the lack of novelty of de novo designs is often criticised[58], hence here we show the nearest known A$_{2A}$ ligand, training set compound, and vendor library compound in the supporting information. Moreover, we test and discover a diverse range of different chemotypes compared to some reported hit rates, which only explore one chemotype[55], provide full dose response curves, a selectivity analysis *versus* A$_{2B}$ receptor, and co-crystallise the two most potent binders to inform future iterations of structure-based design. Moreover, this workflow is practical to implement with each experiment taking 5.94 hours on average using a single consumer-grade GPU (here an NVIDIA RTX 2080Ti) and parallelised over approximately 30 CPUs; therefore, possible to complete overnight.

Despite the use of generative AI, this workflow still uses human expertise to filter and select molecules for testing, resulting in level 1 automated chemical design[59]. Challenges remain in fully automating this workflow and removing human influence while maintaining its success, challenges that are not unique to our approach. We note that visual inspection and novelty analysis for the selection of compounds for testing do not differ greatly between virtual screening and de novo approaches; however, the level of enrichment of A$_{2A}$ chemistry beyond vendor libraries we have shown means that more success is to be found within the top-ranked compounds. One key improvement would be more accurate scoring functions that go beyond providing enrichment to providing reliable, precise predictions for binding affinity and other important molecular properties at reasonable computational expense. As RL is a decision-making algorithm to optimize an arbitrary reward, the resulting decisions (i.e., molecular structures) are only as good as the arbitrary reward is descriptive of ground truth desirability.

## Methods
### Datasets
A dataset of SMILES was used to pre-train the generative model used in this work. This consisted of molecules extracted from ChEMBL28[34], which underwent further filtering and refinement. First, only molecules with a pChEMBL value greater than 6 from an assay with a confidence value of 8 or higher were considered, as a proxy for more developed chemistry resulting in more medicinally interesting chemistry. These molecules were further standardized, neutralized, and filtered to ensure molecules had a predicted logP less than or equal to 4.5, rotatable bond count less than or equal to 7, molecular weight in the range 150 to 650 Da and only contain atoms belonging to the following set A ∈ {C, S, O, N, H, F, Cl, Br}. Any molecules violating MOSES structural alerts were removed[15]. Finally, molecules were clustered based on scaffold similarity using ECFP4 fingerprints at a threshold of 0.8 Tanimoto similarity and only the centroids were

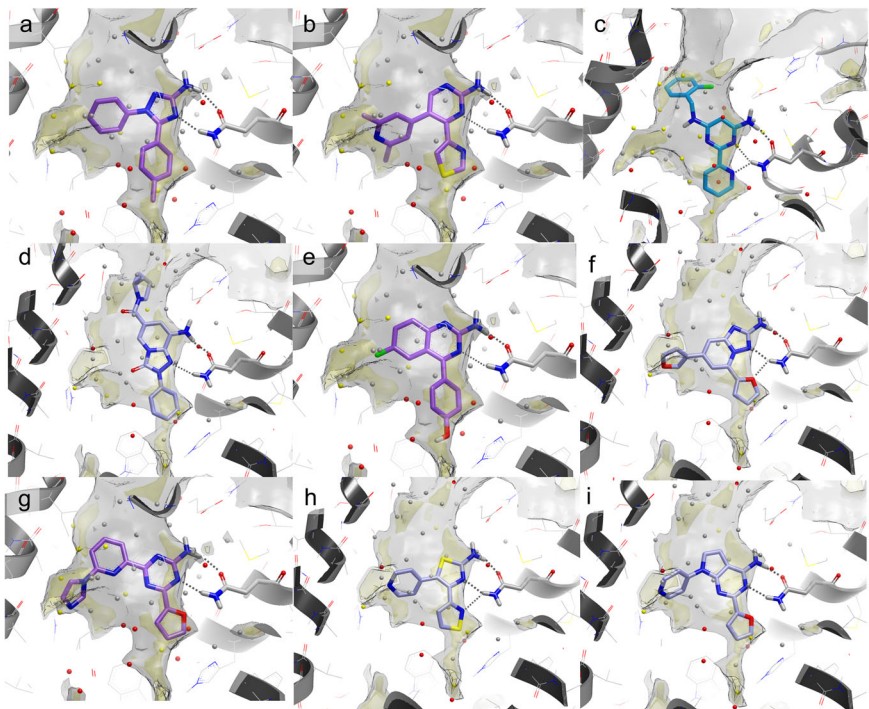

**Fig. 6 | Predicted binding pose of molecules that underwent synthesis and validation.** Docking pose of the nine selected compounds, ordered by compound number 1-9 (from panels a to i). Ligands are coloured according to the crystallographic structure used to retrieve the docking pose (purple carbons for 6GT3, grey carbons for 4EIY and cyan carbons for 5OLH). Other aspects of binding site representation are as in Fig. 4e.

**Table 1 | Summary table showing compound name, binding affinity, and functional activity**

| Compound | $A_{2A}$ | | | | $A_{2B}$ | |
| --- | --- | --- | --- | --- | --- | --- |
| | $pK_i$ (± SD) | $pIC_{50}$ (± SD) | $pEC_{50}$ (± SD) | $E_{max}$ (%± SD) | $pEC_{50}$ | $pIC_{50}$ |
| **1** | 4.8 (±0.3) | Inactive | Inactive | ND | Inactive | Inactive |
| **2** | 5.0 (±0.8) | Inactive | Inactive | ND | Inactive | Inactive |
| **3** | 5.9 (±0.3) | Inactive | Inactive | ND | Inactive | Inactive |
| **4 (Isomer 1)** | 6.4 (±0.0) | 5.3 (±0.4) | Inactive | ND | Inactive | Inactive |
| **4 (Isomer 2)** | 6.2 (±0.1) | 6.5 (±0.4) | Inactive | ND | Inactive | Inactive |
| **5** | 5.7 (±0.1) | Inactive | 7.3 (±0.1) | 65 (±6.8) | Inactive | Inactive |
| **6** | 5.9 (±0.0) | Inactive | Inactive | ND | Inactive | Inactive |
| **7** | 7.3 (±0.1) | 7.6 (±0.1) | 7.7 (±0.2) | −289 (±17.1) | Inactive | Inactive |
| **8** | 4.4 (±0.1) | Inactive | 5.9 (±0.2) | −179 (±3.8) | Inactive | Inactive |
| **9** | 7.5 (±0.1) | 7.7 (±0.1) | 6.5 (±0.1) | −181 (±5.9) | Inactive | Inactive |

[3H]-ZM241385 radioligand competition binding assays for the compounds synthesised were carried out in BacMam transfected $A_{2A}$ cells. Effects of the compounds synthesised on the levels of cAMP measured after overexpression of the wild-type $A_{2A}$, either alone ($pEC_{50}$) or in the presence of an $EC_{80}$ of NECA ($pIC_{50}$). ND refers to not determined. Values correspond to the mean and standard deviation (SD) with $n = 2$ technical replicates for binding assays confirmed by $n = 3$ technical replicates on independent orthogonal agonist and antagonist functional assays.

carried forward into the resulting dataset of 189,238 unique molecules. This under-sampling approach was used to even out the chemical space. During training, 10-fold restricted randomisation of the SMILES representation was conducted to augment the training dataset[60].

Seven $A_{2A}$ receptor crystal structures were selected to represent different possible protein conformations and associated complementary chemistry, six of which are StaRs[61] published in previous work. These included 3REY co-crystallised with XAC[41], 4IEY co-crystallised with ZM241385[62], 5OLO co-crystallised with Tozadenant[63], 5OLH co-crystallised with Vipadenant[63], 5OLV co-crystallised with LUAA47070[63], 6GT3 co-crystallised with HTL1071/AZD4635[64], and 6ZDR co-crystallised with Chromone[65]. For each structure, missing side chain atoms were added, and ionisable residues were set to their most probable protonation state at pH 7.4, and histidine residues in the binding site based on visual inspection. H250[6.52] was protonated in the

Nε position, H278[7.43] was protonated in both Nδ and Nε positions and based on interactions with E169[ECL2] resulting from the conformation of ECL3, H264[ECL3] was either doubly protonated (4EIY[62], 5OLH[66], 5OLV[66], 6GT3[64], 6ZDR[65]) or protonated in the Nδ position (3REY[41], 5OLO[66]). Compounds to dock were prepared using Moka[67,68] followed by Corina[69]. Tautomeric and protomeric states with an abundance of at least 20% at pH 7.4 were enumerated with TauThor[68] and Blabber[67], respectively. Docking was performed using Glide SP[70]. Hydrogen bond constraints were considered to orient the hydrogen bond donor/acceptor-containing scaffolds into sub-pocket I (Fig. 4e), according to hydrogen bonds formed between co-crystallised ligands and N253[6.55] in each structure. An additional occupancy constraint was used with the requirement of a hydrophobic substituent to occupy sub-pocket II (Fig. 4e). All constraints were required to be satisfied after docking, to finally save a top-scored pose for visual inspection.

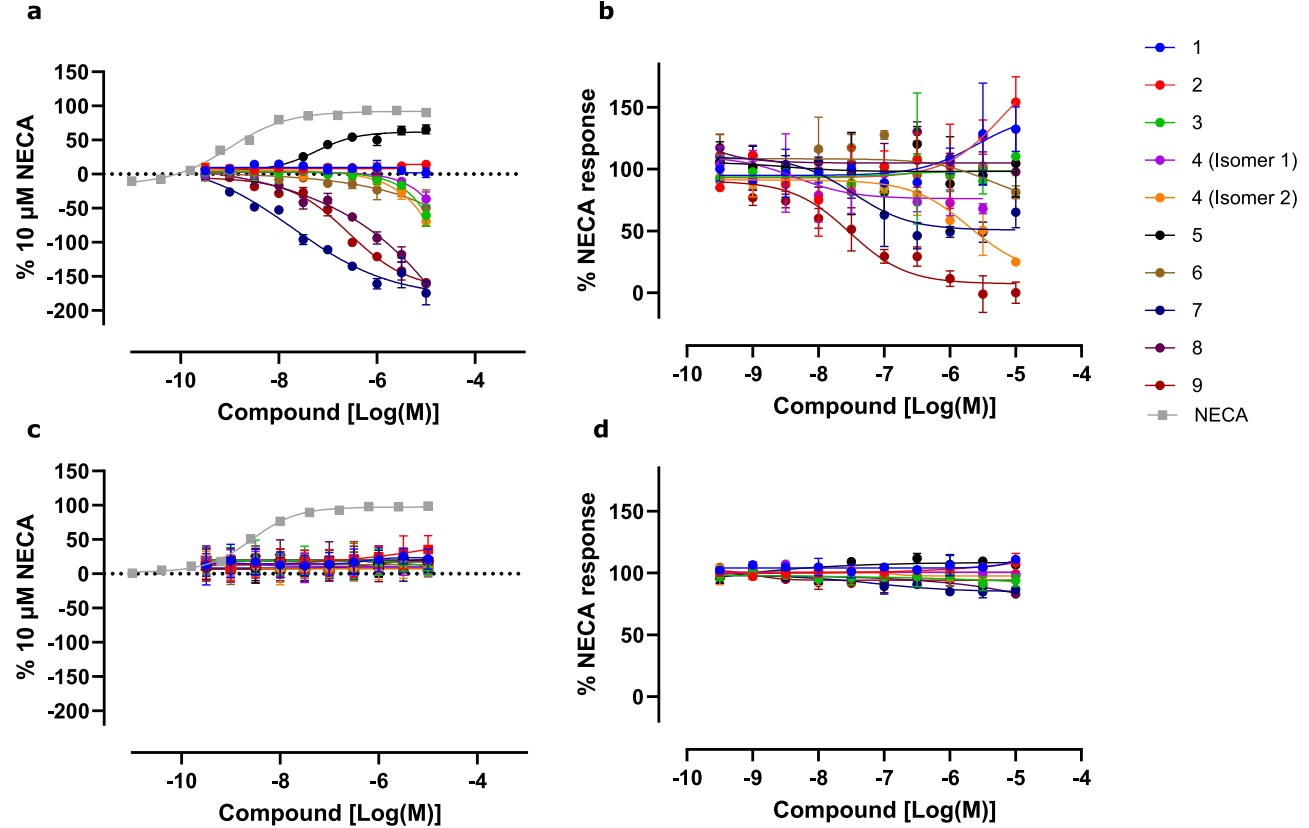

**Fig. 7 | Pharmacological characterisation of proposed A$_{2A}$ ligands.** Compounds of interest were tested in cAMP accumulation assays (Cisbio) in A$_{2A}$ (**a**, **b**) and A$_{2B}$ (**c**, **d**) CHO-K1 cells. Compounds were tested alone to determine agonist activity (a, c) or in the presence of an EC$_{80}$ of NECA (b, d). Overexpression of the wild-type A$_{2A}$ receptor resulted in high levels of constitutive activity. Agonist data was normalised to DMSO control and 10 μM NECA, and was plotted using GraphPad Prism v10, using non-linear regression (four parameter fit). Antagonist data were normalised to assay controls, where 100% response equalled 6 nM NECA (A$_{2A}$) or 19 nM NECA (A$_{2B}$) and 0% response equalled 10 μM literature antagonist AB928 (A$_{2A}$) or GS6201 (A$_{2B}$). **a–d** Three technical replicates (*n* = 3) were conducted with the mean and standard deviation shown. Source data are provided as a Source Data file.

Known A$_{2A}$ chemotypes were taken as previously curated in GPCR-Bench[39], and updated to add more recent chemotypes derived from Reaxys data with a pKi threshold above or equal to 6.6 on human A$_{2A}$ (ligands containing a ribose were discarded, due to likely agonistic functionality.

### Generative model

In this work, a gated recurrent unit (GRU-based) neural network was used as a SMILES-based chemical language model (CLM). These well-detailed models[8,12,13] learn to predict the next token ($x_{i+1}$) in a sequence of tokens ($X$) by training on a corpus of example sequences while using teacher forcing to ensure that the predicted token at each index $y_i$ is calculated with the ground truth previous token $x_i$ as input. The model is then trained to maximize the likelihood assigned to the correct token at each index, conditional upon all previous tokens observed, which is then summed over the whole sequence as formulated in Eq. 1. In this work, SMILES[11] was used as the language-based chemical grammar and the model consisted of an embedding layer of size 256 followed by 3 layers of GRUs with a hidden dimension size of 512. The model was trained on the pre-training dataset described earlier for 5 epochs using a batch size of 128 and the Adam optimizer with a learning rate of 0.001.

$$\mathcal{L}(\theta) = -\sum_{i=o}^{N} logP(x_i|x_{i-1}, \dots, x_0) \qquad (1)$$

Given a trained GRU-based neural network, new SMILES strings can be sampled by inputting a start token ("GO") and iteratively sampling from the predicted probability distribution over the next token and repeating this process by using the sampled token as the next input until a stop token is sampled ("EOS").

### Reinforcement learning

The task of predicting the next token in a sequence can be framed as a partially observable Markov Decision Process, lending itself to the use of RL as an approach to learn which decisions to make in the process to maximize an arbitrary reward. Where each token $x_i$ is considered a state out of all possible states $s \in S$ at a given timestep in a trajectory $t \in \tau$, the transition between two states is determined by an action out of all possible actions $a \in A$. Combining this framework with our generative model, the pre-trained network that learns the probability $P(x_i|x_{i-1}, \dots, x_0)$ can equally be considered an initialized policy where the policy function parameterized by the network weights denotes the probability of taking an action given the state at a given timestep $\pi_\theta(a_t|s_t)$. Within this framework, many different RL algorithms exist to modify the policy, such as to optimize an arbitrary reward $R$. REINFORCE[71] is a policy-based algorithm that only relies on one reward value at the end of the episode (a.k.a return): a particularly favourable property considering a partially complete SMILES string may be invalid or not have a relevant or associated calculable reward. REINVENT[12] is an extension of this algorithm that utilises two networks and therefore two policy functions: a prior that is fixed based on the pre-trained neural network $\pi_{prior}$ and an agent network that is updated to maximize the reward $\pi_{agent}$. The loss function used in REINVENT is shown in Eq. 2, it can be considered a reward shaping of REINFORCE that couples the two policies, such that the agent policy does not drift too

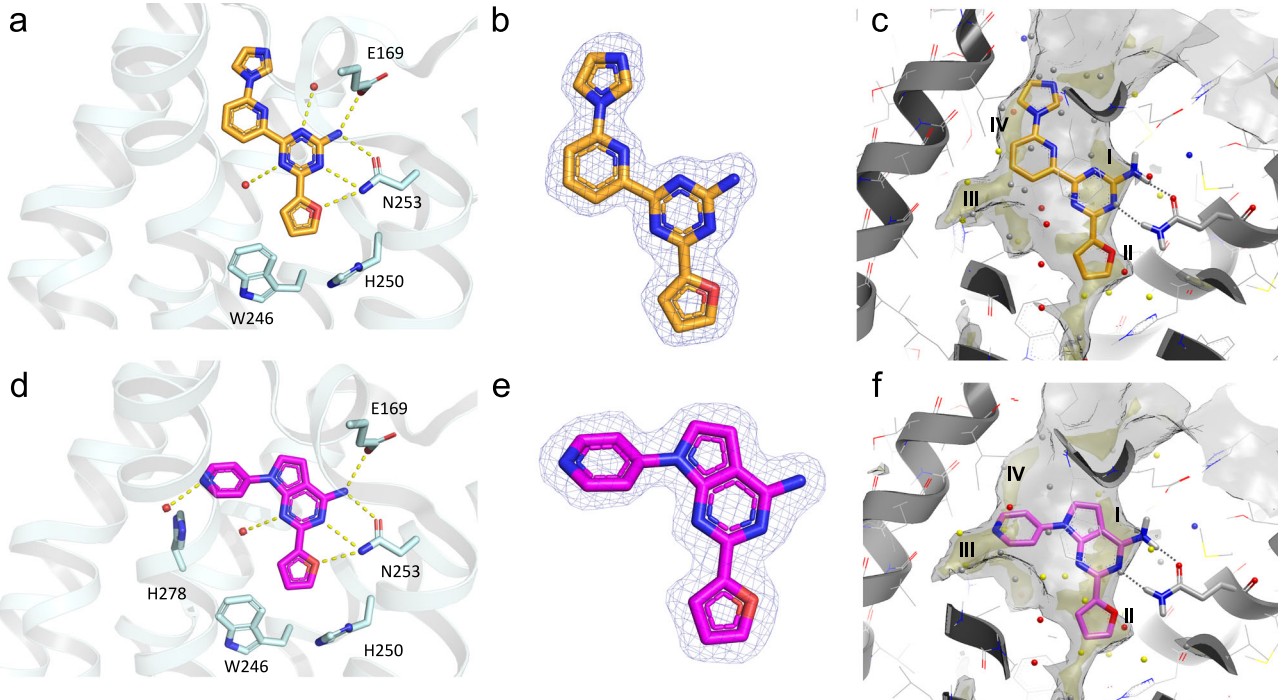

**Fig. 8 | Crystal structure visualisations of Compound 7 (9H2X, orange sticks) and Compound 9 (9H37, purple sticks). a**, **d** The $A_{2A}$-StaR2-$b_{RIL}$ 562-Compound complex shown as a cartoon, with helices coloured in pale cyan. Interesting orthosteric binding site residues and water molecules involved in ligand binding are represented as sticks and as spheres, respectively. Hydrogen bonds are shown as dotted lines. (**b**,**e**) 1.0 σ contoured 2m*Fo*-d*Fc* electron density map corresponding to the ligand represented as a blue mesh, carved around the ligand. (**c**,**f**) The receptor is shown with a cartoon representation and residue side chains as wireframe with grey carbons, with N253[6.55], and the ligand represented as sticks. GRID maps and WaterFLAP water networks on the pseudo-apo site are represented similarly to Fig. 4.

far from the initially learnt prior policy. The extent to which the prior policy is considered in relation to the reward for a given molecule is controlled by σ. This loss function is then used to update the agent such that the probability distribution of actions given a state is learned to maximize the final reward. Augmented Hill-Climb[30] (AHC) modifies REINVENT such that only the top-*k* ranked molecules in a batch are used to update the network weights using Eq. 2 which drastically improves the learning efficiency of the algorithm. Thus, in this work we used AHC to conduct RL with σ = 60, *k* = 0.5, *batch size* = 64 which was conducted for 200 steps, resulting in 12,800 molecules sampled by the agent network.

$$\pi_{aug} = \pi_{prior} - \sigma(R)$$
$$\mathscr{L}(\theta) = \left[\pi_{aug} - \pi_{agent}\right]^2 \qquad (2)$$

**Scoring functions**

Reinforcement learning requires a reward value *R* for a given molecule, in this work between 0 and 1, reflecting the desirability of a particular compound. This was calculated as the arithmetic mean of 5 individual scores $S_x$ between 0-1 describing the desirability of a given molecule with respect to each parameter.

$$f_{maxmin}(x; x_{max}, x_{min}) = \frac{x - x_{min}}{x_{max} - x_{min}} \qquad (3)$$

1. Docking score: Proposed de novo molecules were first prepared to enumerate protonation and tautomerisation states more than 20% abundant at pH 7.4 using MoKa[67], followed by enumeration of unspecified stereocentres up to a total of 16 by CORINA[69].

These variants were then docked using Glide-SP[70]. The lowest (i.e., best) docking score achieved by any molecule variant was returned as the final docking score. To realise the final score $S_{dock}$, the docking score *ds* was normalized based on the maximum and minimum observed docking scores achieved at a given point in training (in reverse, such that a low docking score is given a high score).

$$S_{dock} = f_{maxmin}(ds; \min(ds), \max(ds)) \qquad (4)$$

2. Synthesisability score: Proposed de novo molecules were predicted synthesisable according to RAScore[38], which returns a value between 0 and 1 representing the probability a molecule is predicted to be synthesisable by AiZynthFinder. In this work, the provided XGB model trained on ChEMBL was utilised. As this value is already in the desired range, no transformation was necessary resulting directly in $S_{synth}$.

3. logP: The logP of de novo molecules was predicted using Crippen logP available in RDKit[72,73]. The desirable range specified was between 1 and 3, with a 'hard limit' specified as between 0 and 4. The logP was thus transformed between 0 and 1 according to Eq. 5.

$$S_{\log P} = \begin{cases} 1, & if\ 1 \le \log P \le 3 \\ 0, & if\ \log P \le 0\ or\ \log P \ge 4 \\ f_{maxmin}(\log P; 1, 0), & if\ 0 < \log P < 1 \\ f_{maxmin}(\log P; 3, 4), & if\ 4 > x_i > 3 \end{cases} \qquad (5)$$

4. Rotatable bonds: To restrict the flexibility of the molecule to a sensible range, a consideration of rotatable bonds was included as a scoring component. More specifically, the maximum consecutive number of rotatable bonds present in a de novo

molecule. The desirable range specified was 3 or below, which would result in a score $S_{crot}$ of 1; 4 or more consecutive rotatable bonds resulted in a score of 0.

5. Hydrogen bond donors: The number of hydrogen bond donors was specified as the last scoring component $S_{hbd}$. The desirable range specified was 3 or below, which would result in a score $S_{hbd}$ of 1; 4 or more hydrogen bond donors resulted in a score of 0.

All scoring functions and transformations were implemented as available in MolScore[74].

## Radioligand binding assay
Cell membranes expressing the human $A_{2A}$ receptor were incubated with [³H]-ZM24385 in assay buffer (50 mM HEPES, pH 7.4) in a total assay volume of 200 μl with a final DMSO concentration of 1%. After 120 minutes incubation at room temperature, the reaction was terminated by rapid filtration through GF/C 96-well glass fibre plates with 5 × 0.25 ml washes with ddH₂O using a Tomtec cell harvester. Bound radioactivity was determined through liquid scintillation using Lablogic SafeScint and detected on a MicroBeta liquid scintillation counter. Non-specific binding was determined as that remaining in the presence of a 1 μM saturating concentration of the antagonist CGS15943. Competition binding was performed as above by incubating membranes with 0.5 nM concentration of [³H]-ZM24385 and a range of concentrations of the test compound (10-point concentration curves at half-log intervals). IC₅₀ values were derived from fitting to a four-parameter logistic equation in PRISM (GraphPad Software, San Diego, CA, USA). Apparent Ki values were derived using the equation of Cheng and Prusoff[63].

## Functional assay
**Cell Line Generation.** CHO-K1 cells were BacMam transfected, for 24 hr, with a vector containing human $A_{2A}$ (5% v/v) or $A_{2B}$ (5% v/v). CHO $A_{2A}$ and CHO $A_{2B}$ cells were maintained in Dulbecco's modified Eagle's medium/Ham's F-12 medium (Sigma) supplemented with 10% FBS. Once transfected, cells were harvested and frozen at −150 °C until required for assay.

**cAMP Assays.** CHO $A_{2A}$ and CHO $A_{2B}$ cells were thawed, and seeded at 500 and 1000 cells per well, respectively, in white 384-well plates in assay buffer (Hank's balanced salt solution; Lonza, Basel, Switzerland) supplemented with 100 μM rolipram and 1 U/mL adenosine deaminase, pH 7.4) in the absence or presence of antagonists, for 1 hour at 37 °C. EC₈₀ ($A_{2A}$ 6 nM/$A_{2B}$ 19 nM) concentration of NECA, were added and plates further incubated for 30 minutes. cAMP was detected using cAMP Gs Dynamic kit (Cisbio, Codolet, France) according to the kit instructions. Plates were read on a PHERAstar FS microplate reader (BMG LabTech, Offenburg, Germany) using standard homogeneous time-resolved fluorescence settings. Homogeneous time-resolved fluorescence ratios were determined by dividing emissions at 665 nm by emissions at 620 nm and multiplying by 10,000.

**Data analysis.** The percentage agonist response was normalized to 10 μM NECA (100%) and DMSO (0%). The percentage antagonist response was normalised to $A_{2A}$ and $A_{2B}$ literature antagonist, AB928 or GS6201.

All data was analysed in GraphPad Prism v10 using a non-linear regression four-parameter fit.

## Protein expression and purification
The expression and purification of the $A_{2A}$-StaR2-$b_{RIL}$ 562 construct was carried out following the protocol described previously[41,63]. The receptor was expressed using the Bac-to-Bac Expression System (Invitrogen) in *Trichoplusa ni* Tni PRO cells using ESF 921 medium

(Expression Systems) supplemented with 5% (v/v) foetal bovine serum (Sigma-Aldrich) and 1% (v/v) Penicillin/Streptomycin (PAA Laboratories). Cells were infected at a density of 2.6 × 106 cells/ml with virus at an approximate multiplicity of infection of 1 and grown for 48 hours at 27 °C with constant shaking.

All protein purification steps were carried out at 4 °C unless otherwise stated. Cell pellets were resuspended in 40 mM TRIS buffer at pH 7.6, 1 mM EDTA supplemented with Complete EDTA-free protease inhibitor cocktail tablets (Roche) and disrupted at -15,000 psi using a microfluidizer (Processor M-110L Pneumatic, Microfluidics). Membranes were pelleted by ultra-centrifugation at 200,000 g for 50 minutes, and then subjected to a high salt wash in a buffer containing 40 mM Tris pH 7.6, 1 M NaCl and Complete EDTA-free protease inhibitor cocktail tablets. Washed membranes were resuspended in 50 mL 40 mM Tris pH 7.6 supplemented with Complete EDTA-free protease inhibitor cocktail tablets and 3 mM theophylline (Sigma Aldrich) and incubated for 2 hours at room temperature. Membranes were solubilised with 1.5% n-Decyl-β-D-maltopyranoside (DM, Anatrace), for 2 hours at 4 °C. Solubilised material was centrifuged at 145 000 g for 60 min and supernatant applied to a 5 ml Ni-NTA Superflow cartridge (Qiagen) pre-equilibrated in 40 mM Tris pH 7.4, 200 mM NaCl, 0.15% DM, 1 mM theophylline. The column was washed with 25 column volumes of buffer 40 mM Tris pH 7.4, 200 mM NaCl, 0.15% DM, 70 mM imidazole, 1 mM theophylline and then the protein was eluted with 40 mM Tris pH 7.4, 200 mM NaCl, 0.15% DM, 280 mM imidazole, 1 mM theophylline. Fractions containing $A_{2A}$-StaR2-$b_{RIL}$ 562 were pooled and concentrated using an Amicon Ultra Ultracell 50 K ultrafiltration membrane and applied to a SuperdexTM200 Increase size exclusion column (Cytiva) pre-equilibrated with 40 mM Tris pH 7.4, 200 mM NaCl, 0.15% DM, 1 mM theophylline. Eluted fractions containing the protein were analyzed by SDS PAGE, pooled and concentrated to -35 mg/ml using an Amicon Ultra Ultracell 50 K ultrafiltration membrane and subjected to an ultra-centrifugation at 436 000 g prior to crystallisation.

## Crystallisation, in meso soaking and crystal harvesting
The $A_{2A}$-StaR2-$b_{RIL}$ 562-theophylline co-crystallisation and in *meso* soaking were performed following a well-established protocol[63]. The concentrated protein was mixed using the twin syringe method[75] with monoolein (Nu-Chek) supplemented with 10% (w/w) cholesterol (Sigma Aldrich) and 10 μM theophylline. The final protein-to-lipid ratio was 40:60 (w/w). Forty-nanolitre LCP boli were dispensed onto 96-well Laminex Glass Bases (Molecular Dimensions Ltd.) using a Mosquito LCP crystallisation robot (TTP Labtech) and overlaid with 800 nL precipitant solution. Glass bases were sealed using Laminex Film covers (Molecular Dimensions Ltd). Plate-shaped crystals grew at 20 °C over 2 weeks in 0.1 M tri-sodium citrate pH 5.3–5.4, 0.05 M sodium thiocyanate, 29–32% PEG400, 2% (v/v) 2,5-hexanediol and 0.5 mM theophylline.

For soaking experiments, incisions were made into the Laminex cover over base wells containing crystals. 10 μL of mother liquor containing 1 mM of either Compound 7 or 9 were added to the well, this was then re-sealed using Crystal Clear Sealing Tape (Hampton Research). Crystals were incubated for 1 hour with a final ligand concentration of 925 μM. Single crystals were mounted in LithoLoops (Molecular Dimensions Ltd) and flash-frozen in liquid nitrogen without the addition of further cryoprotectant.

## Diffraction data collection and processing
X-ray diffraction data were collected at beamline I24 (Diamond Light Source) at a wavelength of 0.61992 Å on an Eiger 16 M detector. Crystals were exposed using 80% beam transmission for 0.05 seconds per 0.2° oscillation per frame using an attenuated beam to reduce radiation damage.

**Table 2 | Data collection and refinement statistics for X-ray crystal structures**

| | A$_{2A}$-StaR2-$b_{RIL}$ 562-Compound 7 | A$_{2A}$-StaR2-$b_{RIL}$ 562-Compound 9 |
|---|---|---|
| Data collection | | |
| Space group | C222$_1$ | C222$_1$ |
| Cell dimentions | | |
| a, b, c (Å) | 39.35, 179.53, 140.12 | 39.46, 178.85, 139.70 |
| α, β, γ (°) | 90.0, 90.0, 90.0 | 90.0, 90.0, 90.0 |
| Resolution (Å) | 44.88- 1.75 (1.84-1.75)$^a$ | 44.71-1.72 (1.78-1.72)$^a$ |
| $R_{pim}$ | 0.076 (1.04)$^a$ | 0.060 (1.31)$^a$ |
| I/σ(I) | 11.1 (1.4)$^a$ | 14.0 (1.4)$^a$ |
| CC1/2 | 0.994 (0.540)$^a$ | 0.987 (0.480)$^a$ |
| Completeness (%) | | |
| spherical | 86.4 (27.8)$^a$ | 88.7 (25.3)$^a$ |
| ellipsoidal | 90.0 (38.5)$^a$ | 90.7 (29.9)$^a$ |
| Redundancy | 29.2 (32.3)$^a$ | 30.1 (32.1)$^a$ |
| Refinement | | |
| Resolution (Å) | 44.9-1.75 | 44.7-1.72 |
| $R_{work}$/$R_{free}$ | 20.86/22.64 | 20.47/22.44 |
| No. atoms | | |
| Protein | 3084 | 3077 |
| Ligand | 34 | 32 |
| Other | 447 | 485 |
| B factors | | |
| Protein | 30.04 | 30.76 |
| Ligand | 17.29 | 17.49 |
| Other | 43.85 | 46.46 |
| R.M.S. deviations | | |
| Bond lengths (Å) | 0.008 | 0.008 |
| Bond angles (°) | 0.800 | 0.820 |

Ramachandran plot statistics:
- A$_{2A}$-Compound 7 (residues in Preferred regions: 339/99.12%; Allowed regions: 3/0.88%; Outliers: 0/0.00%).
- A$_{2A}$-Compound 9 (residues in Preferred regions: 346/98.58%; Allowed regions: 5/1.42%; Outliers: 0/0.00%).
$^a$Values in parentheses are for the highest-resolution shell.

Diffraction data from individual crystals were integrated using XDS[76]. Data merging and scaling were carried out using *AIMLESS* (CCP4 suite)[77] and anisotropic correction using STARANISO[78]. The final datasets for both Compound 7 and Compound 9, were merged from 6 sweeps. Data collection statistics are reported in Table 2.

**Structure solution and refinement**
The structures of the different A$_{2A}$-StaR2-$b_{RIL}$ 562-ligand complexes were solved by molecular replacement (MR) with Phaser[79] (CCP4 suite)[77] using the A$_{2A}$-StaR2-$b_{RIL}$ 562-theophylline complex structure as the search model (PDB code: 5MZJ). Iterative cycles of manual model building, and refinement were performed using COOT[80] and Buster[81], respectively. 2 TLS groups corresponding to the receptor + ligand and to the bRIL 562 were defined during refinement. The final refinement statistics are presented in Table 2. Structure figures were generated using PyMOL[82].

**Reporting summary**
Further information on research design is available in the Nature Portfolio Reporting Summary linked to this article.

## Data availability
The crystal structure data of Compound 7 and Compound 9 have been deposited in the PDB under the accession codes 9H2X and 9H37, respectively. The prior training dataset and pre-trained neural network weights are available on GitHub [https://github.com/MorganCThomas/SMILES-RNN] or Zenodo [https://doi.org/10.5281/zenodo.11356193]. Source data are provided with this paper as a Source Data file. Source data are provided with this paper.

## Code availability
All of the code used to generate the results in this work is available open-source under MIT license. The SMILES-RNN[83] repository [https://github.com/MorganCThomas/SMILES-RNN] [https://doi.org/10.5281/zenodo.11356193] contains code to train the generative model and conduct reinforcement learning. The MolScore repository[84] [https://github.com/MorganCThomas/MolScore] [https://doi.org/10.5281/zenodo.14998609] was used to define the objective functions that scored the de novo molecules.

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

## Acknowledgements

The authors acknowledge open-source tools used in this work. MT acknowledges Nxera Pharma UK for their sponsorship of his PhD studies. The permission to publish this work was granted by Nxera Pharma UK.

## Author contributions

M.T. conducted the in silico generative molecular design and analysis with supervision from P.G.M., F.D. and Cd.G. in the preparation and use of crystal structures for docking. M.T., PGM, F.D., R.J.G., J.S.M. and CdG triaged the de novo molecules into those selected for synthesis. R.J.G., K.H. and C.F. were responsible for the synthesis and characterisation of the selected compounds. H.N. and J.B. conducted the binding affinity and pharmacological assay of the synthesis compounds. M.N. and T.G. conducted the crystallographic structure experiments and deposited the structures to the protein data bank (PDB). The project was supervised and managed by F.D., N.A.S., NOB, A.B. and Cd.G. M.T. prepared the manuscript with contributions from P.G.M., H.N., R.J.G., F.D, A.B. and Cd.G. All authors read, edited, and approved the manuscript.

## Competing interests

PGM, RJG, MN, HN, JSM, JB, KH, CF, NAS, TG, NOB, FD, and CdG are currently, or were, employees of Nxera Pharma UK while the work presented in this manuscript was conducted. Nxera Pharma UK is a drug discovery and development company working in the field of G-protein-coupled receptor structure-based drug design. All remaining authors declare no competing interests.
