## [Transparent Peer Review file · Nature Communications]

Identification of nanomolar adenosine A2A receptor ligands using reinforcement learning and structure-based drug design

Corresponding Author: Dr Francesca Deflorian

Version 0:

Reviewer comments:

Reviewer #1

(Remarks to the Author)

This is an interesting and well written report on using deep learning tools to discover novel adenosine A2AR ligands featuring new chemotypes. The most promising virtual hit compounds were synthesised and experimentally evaluated, thereby yielding novel A2AR ligands with potent and useful properties. Two of the active compounds were then also co-crystallised with A2AR and compared to the predicted docking poses. I think this work is highly relevant to discover novel chemotypes in the area of adenosine receptors, but will also attract great interest in the structure-based drug design field in general, addressing other protein targets. Furthermore, I think that readers not very familiar with deep learning methods are also able to follow the process and "journey" of the compounds, i.e. how the authors arrived at the hit compounds, how they were selected and experimentally validated. Regarding the methodology, I find the methods used to assess the binding and functional activity of the compounds sound and appropriate.

Major comments:

1. Do these nine synthesised compounds also bind and functionally modulate adenosine A1 and and A3 receptors? As the compounds target the orthosteric site of the A2AR, and given that the orthosteric binding sites of the four adenosine receptor subtypes are fairly similar, it would be very informative to see experimentally, if the hit compounds are indeed highly selective for only the A2A subtype.

2. The structural A2AR templates used as starting points in this study (3REY, 4EIY, 5OLO, 5OLH, 5OLV, 6GT3, 6ZDR) all represent receptors bound to antagonists (according to the ligand classification in the IUPHAR/BPS database, although some of these ligands were later re-classified as inverse agonists). I therefore assume that the receptor is mostly stabilised in an inactive conformation by these ligands. How much bias does the choice of structural templates, i.e. receptor conformations, imprint on the outcome of functional properties of the discovered ligands? I have noticed that the synthetic, active compounds were characterised as inverse agonist or antagonists. Would the authors expect to discover full agonists as well, or would one need to use agonist-bound A2AR structural entry templates to get agonist hits?

Minor comment:

It would be useful to provide ¹³C NMR spectra/data for the synthetic final compounds as well.

(Remarks on code availability)

I have only checked that I can access the code and e.g. README files. However, I don't have the expertise to review and judge the code itself.

Reviewer #2

(Remarks to the Author)

The authors combine structure-based drug design (SBDD) principles with chemical language models (CLMs), reinforcement learning (RL), and Augmented Hill-Climb (AHC), and present what they qualified of "modern hit-finding workflow" to go from protein structure to novel small-molecule ligands. Using A2A knowledge, X-ray structures, they identify new chemotypes of the human A2A receptor, predicted 41 synthesizable de novo molecules. Nine were selected, synthesized, and tested for their binding affinity and functional potency. Here two high affinity inverse antagonist, with new chemotypes (it says three in the abstract), are presented and their binding mode revealed by solving high resolution X-ray structures.

Exploring structure-based drug design is an important task, with a growing number of structures now available and new powerful tools (AI). It is important to propose new strategy for exploring chemical space and identifying molecules that can fit into protein binding site.

This article takes a very technical angle on drug screening and discovery, not always accessible for non-specialist in regards of the methodology used for identifying de novo molecules. However, it is interesting to see that the authors identify new molecules, not available in commercial library, for a receptor (A2A) that has been extensively investigated during the last decades.

Line 48-50, what is the meaning of "molecular optimization"?

Line 200, The reference cited here might be the wrong one, ref 38 would fit better here.

Line 202-209. Outcomes of "classical" in silico screening SBDD strategies were strongly influence by the conformational state trapped in the X-ray structure used, i.e if one used an antagonist-bound conformation, top hits are antagonists; and it would be the opposite way for agonists. It is interesting to note that here compound 5 is an agonist, even if partial. Could the authors comment on this point. In addition the chemotype is rather unusual for an A2A agonist?

Here the authors explain that the A2A has high constitutive activity. Constitutive activity is related to receptor expression, meaning that a lower expression level the constitutive activity will be reduce. There is no expression level measured here. Is it really necessary to specify the constitutive activity here? I would rather see a full activation curve for NECA in order for the reader to appreciate the inverse-agonism and agonism effects of the tested molecules relative to a full agonist. The figure (7) is incomplete and difficult to read as it is. In addition, this should allow to include statistical analysis that are missing at the moment. Where is the vehicle point for each curve? Also, please provide ligand binding curve as supplementary figures.

Overall, this is interesting paper, but that might better fit in a more specialized journal.

(Remarks on code availability)

Reviewer #3

(Remarks to the Author)

Thomas and co-authors propose a 'structure-guided' generative pipeline for de novo design, which is validated in the wet-lab. The paper is a nice example of how chemical language modeling approaches can be used to chart unexplored regions in the chemical space. It is also nice to see an experimental validation of reinforcement learning pipelines. Finally, the paper provides a nice, interdisciplinary story on how to narrow down the number of designs proposed by generative models.

In general, the paper is well-written and scientifically-sound. However, I believe a few aspects will increase its impact and clarity to the interdisciplinary readership of Nat Comms.

Major comments:

1. The usage of the word 'structure-guided'. This choice could be misleading, as one might assume that structure-conditioned generation was employed. The structure is used only indirectly -- by using docking scores as reward -- and not explicitly incorporated into the model. I suggest the authors to rephrase, to better match the paper content (e.g., docking guided).
2. Many of the considerations of this paper are based on the expression 'chemotype', which can be interpreted (slightly) differently by every person. Consider defining how you obtain chemotypes at the first occurrence of the term.
3. Several manual steps (as also observed by the authors) allowed to obtain these results. This is not bad per se, but I find that these steps are not always fully explained, which reduces the transparency and reproducibility. For instance, row 172 (p. 7): molecules were inspected and selected based on the predicted binding pose -- how? What were you considering for the choice? Row 182 (p. 8): 41 molecules were proposed for the synthesis and 9 were synthesized based on synthetic route and diversity. What elements of the synthetic route were considered? And what aspects were considered for diversity? It would be good to make every step as reproducible as possible, to help the community build on these nice results.

Minor comments:

1. Results. The RL setup should be better explained here, so that readers do not have to search for information in the materials and method section. The description of figure 2 and the corresponding considerations happen almost abruptly.
2. Consider rephrasing 'Bemis-Murcko scaffold Tanimoto similarity' for clarity
3. Table 1: something seems to have happened with the standard deviation on pKi for compound 4 (i1) and 6.
4. Typo in row 393 (p. 17): synthesi(s)able

(Remarks on code availability)

Version 1:

Reviewer comments:

Reviewer #1

(Remarks to the Author)

I feel that my comments have been adequately addressed in the revised version of the manuscript and recommend publication.

(Remarks on code availability)

Reviewer #2

(Remarks to the Author)

Authors address all my comments.

There is however still one point that is not clear to me regarding figure 7:

In figure legend, line 245-246: "Antagonist data (n=3) was normalised to 100% EC80 response (A2A 6 nM, A2B 19 nM) and 0% 10 μ M literature antagonist AB928 or GS6201."

Here the normalisation is done using EC80 of NECA?, since authors look for inhibition. However, 6nM is far to be the EC80 of NECA? This probably needs to be rephrased, it is difficult to understand how the normalisation was done as it is.

(Remarks on code availability)

Reviewer #3

(Remarks to the Author)

I thank the authors for addressing my concerns.

(Remarks on code availability)

Foremost we would like to thank all the reviewers for the valuable time reviewing our manuscript and providing constructive comments on our research. Here we provide a point-by-point response to the reviewer's comments in black, with our comments *in italics*.

Response to Reviewer #1

This is an interesting and well written report on using deep learning tools to discover novel adenosine A2AR ligands featuring new chemotypes. The most promising virtual hit compounds were synthesised and experimentally evaluated, thereby yielding novel A2AR ligands with potent and useful properties. Two of the active compounds were then also co-crystallised with A2AR and compared to the predicted docking poses. I think this work is highly relevant to discover novel chemotypes in the area of adenosine receptors, but will also attract great interest in the structure-based drug design field in general, addressing other protein targets. Furthermore, I think that readers not very familiar with deep learning methods are also able to follow the process and "journey" of the compounds, i.e. how the authors arrived at the hit compounds, how they were selected and experimentally validated. Regarding the methodology, I find the methods used to assess the binding and functional activity of the compounds sound and appropriate.

The authors appreciate the positive view of the referee regarding the overall aim, methods employed, as well as presentation of the material in the manuscript.

Major comments:

- Do these nine synthesised compounds also bind and functionally modulate adenosine A1 and A3 receptors? As the compounds target the orthosteric site of the A2AR, and given that the orthosteric binding sites of the four adenosine receptor subtypes are fairly similar, it would be very informative to see experimentally, if the hit compounds are indeed highly selective for only the A2A subtype.

The objective of this study was the validation of a de novo design workflow to identify novel A_{2A} adenosine receptor ligands, which we did experimentally validate extensively, as acknowledged by the reviewers. We furthermore want to emphasise that we have determined that the experimentally confirmed hits are all selective for A_{2A} adenosine versus the A_{2B} adenosine receptor subtype as shown in Table 1 and Figure 7 in the manuscript. While in a drug discovery campaign selectivity is of importance, it is not within the scope of our study here.

- The structural A2AR templates used as starting points in this study (3REY, 4EIY, 5OLO, 5OLH, 5OLV, 6GT3, 6ZDR) all represent receptors bound to antagonists (according to the ligand classification in the IUPHAR/BPS database, although some of these ligands were later re-classified as inverse agonists). I therefore assume that the receptor is mostly stabilised in an inactive conformation by these ligands. How much bias does the choice of structural templates, i.e. receptor conformations, imprint on the outcome of functional properties of the discovered ligands? I have noticed that the synthetic, active compounds were characterised as inverse agonist or antagonists. Would the authors expect to discover full agonists as well, or would one need to use agonist-bound A2AR structural entry templates to get agonist hits?

As suggested by the reviewer, we expect more agonists would be discovered when using active state receptor structures. However, we note the caveats that 1) docking is an imperfect prediction of binding mode and affinity, and 2) the differences in protein-ligand interactions between active and inactive states can sometimes be subtle (Ballante, 2021). Therefore, it is not unlikely to

discover alternative functions than supposed (Rodríguez, 2015), as happens in our discovery of a partial agonist despite the use of inactive receptor states.

A paragraph has been added discussing this in the main body of the manuscript, which reads as follows (page 14):

“We note that although 3/4 molecules synthesised functionally inactivated the A_{2A} receptor, one activated the receptor as a partial agonist despite an attempt to bias ligand chemistry using inactive state A_{2A} receptor complexes. However, active and inactive state binding site conformations are generally quite similar, and ligands typically have some affinity for both conformations⁴⁵ therefore it is not surprising to identify ligands with alternative functional effects than expected (especially considering imperfect binding mode prediction with docking). Moreover, a previous study by Rodríguez et al. using activate state A_{2A} receptors in the hunt for agonists only identified antagonists⁴⁷, emphasising the challenge of functional activity prediction. Despite this, we cautiously expect that using active state A_{2A} structures in this workflow would bias ligands more towards agonists because the generative model probes novel chemical space, whereas a chemical bias towards antagonists within the commercial library was identified as part responsible by Rodríguez et al. We also identify scope to integrate protein-ligand interaction fingerprints within the scoring function to predict functional activity⁴⁸ in future iterations.”

Minor comment:

- It would be useful to provide 13C NMR spectra/data for the synthetic final compounds as well.

Only the ¹H NMR spectra was collected as we report and therefore, we do not have this data to include. This is consistent with general medicinal chemistry journal guidelines (such as e.g. the Journal of Medicinal Chemistry (https://researcher-resources.acs.org/publish/author_guidelines?coden=jmcmar) which only require ¹H NMR spectra for compound characterisation.

Response to Reviewer #2

The authors combine structure-based drug design (SBDD) principles with chemical language models (CLMs), reinforcement learning (RL), and Augmented Hill-Climb (AHC), and present what they qualified of “modern hit-finding workflow” to go from protein structure to novel small-molecule ligands. Using A_{2A} knowledge, X-ray structures, they identify new chemotypes of the human A_{2A} receptor, predicted 41 synthesisable de novo molecules. Nine were selected, synthesized, and tested for their binding affinity and functional potency. Here two high affinity inverse antagonist, with new chemotypes (it says three in the abstract), are presented and their binding mode revealed by solving high resolution X-ray structures.

Exploring structure-based drug design is an important task, with a growing number of structures now available and new powerful tools (AI). It is important to propose new strategy for exploring chemical space and identifying molecules that can fit into protein binding site.

This article takes a very technical angle on drug screening and discovery, not always accessible for non-specialist in regards of the methodology used for identifying de novo molecules. However, it is interesting to see that the authors identify news molecules, not available in commercial library, for a receptor (A_{2A}) that has been extensively investigated during the last decades.

Comments:

- Line 48-50, what is the meaning of “molecular optimization”?

This has been clarified with the following text “learning efficiency of the algorithm (i.e., how many samples are required)”.

- Line 200, The reference cited here might be the wrong one, ref 38 would fit better here.

This has been corrected.

- Line 202-209. Outcomes of "classical" in silico screening SBDD strategies were strongly influence by the conformational state trapped in the X-ray structure used, i.e if one used an antagonist-bound conformation, top hits are antagonists; and it would be the opposite way for agonists. It is interesting to note that here compound 5 is an agonist, even if partial. Could the authors comment on this point. In addition, the chemotype is rather unusual for an A_{2A} agonist?

In practice the difference in binding site conformation between inactive and active states can be quite similar (Ballante, 2021) and hence it is not surprising to identify a partial agonist using inactive structures. Previous studies (Rodríguez, 2015) have identified 9 antagonists whilst screening using the active structure.

A paragraph has been added discussing this (combined with a previous comment for Reviewer #1) in the main body of the manuscript, which reads as follows (page 14) :

“We note that although 3/4 molecules synthesised functionally inactivated the A_{2A} receptor, one activated the receptor as a partial agonist despite an attempt to bias ligand chemistry using inactive state A_{2A} receptor complexes. However, active and inactive state binding site conformations are generally quite similar, and ligands typically have some affinity for both conformations⁴⁵ therefore it is not surprising to identify ligands with alternative functional effects than expected (especially considering imperfect binding mode prediction with docking). Moreover, a previous study by Rodríguez et al. using activate state A_{2A} receptors in the hunt for agonists only identified antagonists⁴⁷, emphasising the challenge of functional activity prediction. Despite this, we cautiously expect that using active state A_{2A} structures in this workflow would bias ligands more towards agonists because the generative model probes novel chemical space, whereas a chemical bias towards antagonists within the commercial library was identified as part responsible by Rodríguez et al. We also identify scope to integrate protein-ligand interaction fingerprints within the scoring function to predict functional activity⁴⁸ in future iterations.”

- Here the authors explain that the A_{2A} has high constitutive activity. Constitutive activity is related to receptor expression, meaning that a lower expression level the constitutive activity will be reduce. There is no expression level measured here. Is it really necessary to specify the constitutive activity here? I would rather see a full activation curve for NECA in order for the reader to appreciate the inverse-agonism and agonism effects of the tested molecules relative to a full agonist. The figure (7) is incomplete and difficult to read as it is. In addition, this should allow to include statistical analysis that are missing at the moment. Where is the vehicle point for each curve? Also, please provide ligand binding curve as supplementary figures.

We thank the reviewer for their feedback on the presentation of the experimental pharmacological validation studies, and we have addressed their comments in the following manner:

- The constitutive activity of the receptor was highlighted as some of the compounds appear to be inverse agonists. We have followed the feedback from the reviewer and included a full NECA curve in Figure 7a and c.

- Please note that we have included a statistical analysis in Table 1, providing SD values for pK_i , pIC_{50} and pEC_{50} , based on $n=3$, as indicated in the caption of Table 1. Following the feedback from the reviewer, we have furthermore added SD values for the E_{max} data to Table 1.

- Regarding the proposal from the reviewer to show vehicle points in each curve: We want to emphasise that all curves were normalised to DMSO/ vehicle control, as described in the figure legend.

We have followed the suggestion from the review and included ligand binding curves as Supplementary Figure 5.

- Overall, this is interesting paper, but that might better fit in a more specialized journal.

We would like to reiterate that due to the interdisciplinary nature of the work (as also highlighted by reviewer #3) we believe that the readership of Nat Comms will be find this work highly relevant. Moreover, as stated by Reviewer #1 the manuscript is written to be accessible to a broad audience "I think that readers not very familiar with deep learning methods are also able to follow the process and "journey" of the compounds, i.e. how the authors arrived at the hit compounds, how they were selected and experimentally validated."

Response to Reviewer #3:

Thomas and co-authors propose a 'structure-guided' generative pipeline for de novo design, which is validated in the wet-lab. The paper is a nice example of how chemical language modeling approaches can be used to chart unexplored regions in the chemical space. It is also nice to see an experimental validation of reinforcement learning pipelines. Finally, the paper provides a nice, interdisciplinary story on how to narrow down the number of designs proposed by generative models.

In general, the paper is well-written and scientifically-sound. However, I believe a few aspects will increase its impact and clarity to the interdisciplinary readership of Nat Comms.

The authors appreciate the positive comments of the referee on both contents and presentation of the work, in line with the comments above.

Major comments:

1. The usage of the word 'structure-guided'. This choice could be misleading, as one might assume that structure-conditioned generation was employed. The structure is used only indirectly -- by using docking scores as reward -- and not explicitly incorporated into the model. I suggest the authors to rephrase, to better match the paper content (e.g., docking guided).

Considering this point, we have changed the title to 'structure-based'. We understand the reviewer's perspective, however, also considering the diverse readership of Nat Comms we believe 'structure-based' will be more appropriate. Moreover, this word has been associated with techniques such as docking for many years already. We have updated the title and any references from 'structure-guided' to 'structure-based' throughout.

2. Many of the considerations of this paper are based on the expression 'chemotype', which can be interpreted (slightly) differently by every person. Consider defining how you obtain chemotypes at the first occurrence of the term.

Chemotypes were defined in a previous publication (Weiss, 2016) and the following text has been added on page 4 to clarify that "Note that A2A receptor ligand chemotypes were defined manually by in-house project teams utilising X-ray and docked co-structures as per GPCRBench³⁶".

3. Several manual steps (as also observed by the authors) allowed to obtain these results. This is not bad per se, but I find that these steps are not always fully explained, which reduces the transparency and reproducibility. For instance, row 172 (p. 7): molecules were inspected and selected based on the predicted binding pose -- how? What were you considering for the choice? Row 182 (p. 8): 41 molecules were proposed for the synthesis and 9 were synthesized based on synthetic route and diversity. What elements of the synthetic route were considered? And what aspects were considered for diversity? It would be good to make every step as reproducible as possible, to help the community build on these nice results.

We amended/added the following text to address the comments and feedback from the reviewer:

Page 8: 'Molecules were inspected and selected based on the predicted binding pose (Figure 6). Selection criteria included: i) H-bond interaction with N253^{6.55}; ii) interaction with at least two of the following three lipophilic hotspots II, III, IV, displacing energetically unfavourable unhappy water molecules located in these binding site regions (Figure 4).

Page 8: '(...) 41 molecules were proposed for the synthesis and 9 were synthesized based on synthetic feasibility and diversity. Synthetic chemistry routes of this triaged set of compounds were defined based on organic chemistry knowledge and established synthesis protocols. Chemical diversity-based selection was guided by ECFP4 fingerprint clustering.'

Minor comments:

1. Results. The RL setup should be better explained here, so that readers do not have to search for information in the materials and method section. The description of figure 2 and the corresponding considerations happen almost abruptly.

An additional paragraph has been added to Results on page 3, further explaining the generative model and RL setup over 2 paragraphs which we hope serves as a smoother introduction to the section as follows:

"To generate de novo molecules with optimal properties, we used a CLM trained using RL. First, a recurrent neural network was trained at next token prediction using maximum likelihood

estimation on 189,238 SMILES string extracted from the ChEMBL database³⁴, constituting our CLM. Second, this CLM underwent further training using RL. In RL, the CLM equates to a policy that decides which action (next token) to take given the current state (previously observed tokens), such as to learn how to maximise a reward. We used the AHC³⁰ for its improved learning efficiency compared to baseline algorithms^{12,35–37} which fine-tunes the CLM to maximise a reward bound between [0, 1]. Note that a copy of the pre-trained CLM is kept during RL and is used as a prior policy to regularise learning and maintain the chemical principles initially learned.

AHC was then used to train the CLM to generate molecules optimal against each of the seven A_{2A} receptor crystal structures over the course of sampling 12,800 de novo molecules per structure. The reward maximised was formulated to reflect molecular desirability by combining the predicted protein complementarity according to the GlideSP docking score and four secondary objectives to encourage more favourable drug-like properties. These secondary objectives included synthesisability³⁸, predicted logP, hydrogen bond donor count, and the maximum number of consecutive rotatable bonds present, thus presenting a more realistic and challenging multi-objective optimisation problem.”

2. Consider rephrasing 'Bemis-Murcko scaffold Tanimoto similarity' for clarity

This has been clarified (page 6) by rephrasing to “The unknown chemotypes were clustered using the Tanimoto similarity (using ECFP4 fingerprints) of the Bemis-Murcko scaffolds with a similarity cut-off of 0.2”.

3. Table 1: something seems to have happened with the standard deviation on pK_i for compound 4 (i1) and 6.

We thank the reviewer for their feedback, and we have resolved this in Table 1:

- "0.0" instead of just "0" for compounds 4 and 6
- added brackets in the pK_i column for consistency
- one decimal digit in the pIC₅₀ column for consistency

4. Typo in row 393 (p. 17): synthesi(s)able

Resolved.

References

Ballante, F., Kooistra, A. J., Kampen, S., de Graaf, C. & Carlsson, J. Structure-Based Virtual Screening for Ligands of G Protein–Coupled Receptors: What Can Molecular Docking Do for You? *Pharmacol Rev* 73, 1698–1736 (2021).

Rodríguez, D., Gao, Z. G., Moss, S. M., Jacobson, K. A. & Carlsson, J. Molecular docking screening using agonist-bound GPCR structures: Probing the A_{2A} adenosine receptor. *J Chem Inf Model* 55, 550–563 (2015).

Weiss, D. R., Bortolato, A., Tehan, B. & Mason, J. S. GPCR-Bench: A Benchmarking Set and Practitioners’ Guide for G Protein-Coupled Receptor Docking. *J Chem Inf Model* 56, 642–651 (2016).

We would like to thank all the reviewers for their valuable feedback, we are grateful that we could address many of their concerns in our revised version. Here we provide a response to the last reviewer comment in black, with our comments in *italics*.

Response to Reviewer #2

There is however still one point that is not clear to me regarding figure 7:

In figure legend, line 245-246: "Antagonist data (n=3) was normalised to 100% EC80 response (A_{2A} 6 nM, A_{2B} 19 nM) and 0% 10 µM literature antagonist AB928 or GS6201." Here the normalisation is done using EC80 of NECA?, since authors look for inhibition. However, 6nM is far to be the EC80 of NECA? This probably needs to be rephrased, it is difficult to understand how the normalisation was done as it is.

To clarify the normalization of assay data we have rephrased the sentence which now reads: "Antagonist data (n=3) was normalised to assay controls, where 100% response equalled 6 nM NECA (A_{2A}) or 19 nM NECA (A_{2B}) and 0% response equalled 10 µM literature antagonist AB928 (A_{2A}) or GS6201 (A_{2B})." In addition, we have simplified the y-axis title of Figure 7b,d that now reads "% NECA response" to aid quick understanding (with the additional information in the legend). With regard to the NECA EC80, our reported value of 6 nM lies within the range of values reported in the literature which can vary from 4 nM (Vlachodimou, 2022) to 600 nM (Claff, 2024).

References

Claff, Tobias, et al. "Structural Insights into Partial Activation of the Prototypic G Protein-Coupled Adenosine A_{2A} Receptor." *ACS pharmacology & translational science* 7.5 (2024): 1415-1425.

Vlachodimou, Anna, et al. "Kinetic profiling and functional characterization of 8-phenylxanthine derivatives as A_{2B} adenosine receptor antagonists." *Biochemical pharmacology* 200 (2022): 115027.